# A Scoping Review of Instruments Used in Measuring Social Support among Refugees in Resettlement

**DOI:** 10.3390/ijerph21060805

**Published:** 2024-06-20

**Authors:** Godfred O. Boateng, Karin Wachter, Roseanne C. Schuster, Tanya L. Burgess, Mary Bunn

**Affiliations:** 1School of Global Health, York University, Toronto, ON M3J 1P3, Canada; 2Dahdaleh Institute for Global Health Research, York University, Toronto, ON M3J 1P3, Canada; 3School of Social Work, Arizona State University, Phoenix, AZ 85004, USA; karin.wachter@asu.edu (K.W.); tlburge4@asu.edu (T.L.B.); 4School of Human Evolution and Social Change, Arizona State University, Phoenix, AZ 85069, USA; roseanne.schuster@asu.edu; 5Department of Psychiatry, University of Illinois Chicago, Chicago, IL 60612, USA; mbunn@uic.edu

**Keywords:** social support, refugee, resettlement, measurement, validity, reliability

## Abstract

This study aimed to systematically review current research on the application of existing social support scales in research with refugees in resettlement, assess their quality, and identify gaps in measurement to enhance research and practice. A scoping review was conducted on the extant literature published until March 2023. A team of researchers conducted search, sorting, and data extraction processes following best practices for scale development and validation. Of the 1185 studies collected from the search process, 41 articles were retained in the final analysis, from which 17 distinct social support instruments used in research with resettled refugees were identified. An assessment of all 17 instruments showed the presence of one or more limitations associated with construct, criterion, convergent, and/or discriminant validity. Test of reliability was assessed in all studies, with a range of 0.80 to 0.90. Our findings show that most of the research evaluating social support among resettled refugees is conducted without measurement instruments adequately validated in the resettlement context. This analysis highlights the need for rigorously developed social support scales that reflect the lived experiences, needs, and priorities of resettled refugees.

## 1. Introduction

The meaning of social support is highly contextual [1]. Yet, existing measurement instruments have not reflected social support priorities and needs following war, forced displacement, and refugee resettlement, leading to incongruences between lived experience and measurement, missed opportunities for intervention, and furthering empirical research [2]. While social support is a well-developed concept across disciplines, its application has not always been consistent with the population and context for which it was originally developed. Therefore, the conclusions drawn from studies that use existing scale items without rigorous cultural- and contextual-specific adaptations or psychometric evaluation raise questions regarding validity and interpretation. In general, social support scales do not lend themselves to the universal application of an entire population or subpopulation. To this point, a variety of social support scales have been developed for university students [3], adolescents [4], children [5], families [6], friends [7,8], gay men [9], patients [10], caregivers [11], breastfeeding women [12], and older adults [13], among others. Nevertheless, most social support scales used in research with refugees in resettlement were neither originally developed with resettled refugee populations nor rigorously evaluated or adapted for resettlement. It is our position that the social support needs of refugees who have experienced war, forced displacement, and resettlement differ in distinct ways from the common populations, for which social support scales were originally developed. Therefore, this paper seeks to examine the current application of social support scales in research with refugees in resettlement, assess their quality, and identify gaps in measurement to enhance research and practice.

### 1.1. Social Support Measurement

Social support is a complex and often muddled concept [2]. Although inter-related, social support is distinct from the constructs of social networks and social integration [14]. Definitions of social support include the following: a function of network membership and mutual obligation that cultivates the belief that one is loved and valued, interpersonal transactions, and resources provided by other persons [15,16,17,18]. Cohen, Gottlieb, and Underwood defined social support as informal helping behaviors, specifically referring to it as “the social resources that persons perceive to be available or that are actually provided to them by nonprofessionals in the context of both formal support groups and informal helping relationships” [18]. Conceptualizations of social support typically distinguish the assistance available through one’s informal networks of family, friends, and acquaintances from the help available through formal networks of helping professionals. As a multidimensional construct, supportive functions may include emotional, practical, instrumental, informational, companionship, and esteem support, perceived or received from various levels of the social ecology [19]. Gottlieb and Bergen distinguished perceived versus received support as “…essentially the belief or faith that support is available from network members, whereas actual [received] support is its mobilization and expression” [14] (p. 512).

In 2001, Cohen, Underwood, and Gottlieb strove to bring clarity to social support measurement by differentiating between measuring social integration and social networks, perceived and received social support, relationship properties and interactions relevant to social support, and by elucidating the perspectives and theoretical roots of those approaches (e.g., stress and coping theory, experimental social psychology, pragmatic philosophy, and sociology) [18]. Stress and coping theory emphasizes the association between received support and coping, which in turn buffers (moderates) the relationship between stress and health, or perceptions of how the availability of specific types of support shapes appraisals of stressful situations, and thus buffers or moderates the effects of stress on health and well-being [18,20,21].

Periodic reviews have examined the evolution of how social support has been measured, the application of these measures, and the shortcomings of existing measures in use with specific populations. In 2001, Wills and Shinar did an in-depth review of standardized social support instruments [22]. Their compilation of measures of perceived social support totaled 21 instruments, categorized as brief unidimensional measures; brief compound measures; broad scales of close and diffuse support; multidimensional inventories; network-based inventories; and measures for children and adolescents. Their list of received social support measures, in contrast, had five in total, four of which were also listed as measures of perceived support. With regards to adaptation to specific populations, Wills and Shinar noted that “...research on social support and ethnicity is less developed. While evidence of cross-cultural validity is available, some studies have suggested caution about using current measures of social integration or social support in other cultures” [22]. The authors went on to suggest that “researchers working in other cultures may need to do more pilot studies to test the relevance of developed scales and extend understanding of how supportive relationship are conceptualized in a given cultural population” [18,22]. Later, in 2010, Gottlieb and Bergen defined social support and related constructs and reviewed observational and self-reported measures of social support, in which they distinguished between “generic and specialized measures” and provided an in-depth overview of three scales, including their psychometric properties [14]. In 2019, a systematic review identified 19 social support scales utilized among HIV-positive/-affected populations, the majority of which underwent limited psychometric testing [23]. A 2024 systematic review of social support instruments in the treatment of post-traumatic stress disorder (PTSD) identified and assessed 29 instruments, 14 of which were assessed as having sufficient content validity and reliability; 9 instruments were assessed as having comparable properties in samples with PTSD [24]. These reviews highlight variations in social support measurement tools and the extent to which they are often applied to people and situations, for which they were not originally developed.

### 1.2. Social Support among Refugees in Resettlement

Culture and context, as well as intersectional social locations and identities, shape meanings, expressions, and expectations of social support [1,2,25]. It is thus important to situate people (refugees) undergoing resettlement in relation to their lived experiences before and following resettlement. People with refugee and similar statuses (e.g., humanitarian parolee, asylee) and forced migrants who do not have the relative protection of an official status have, on the whole, experienced horrors associated with war, persecution, violence, forced displacement, and immeasurable losses of and separations from loved ones, resources, ways of life, home, and country [26]. These traumatic experiences can erode people’s capacities to attach securely, trust others, and share emotional intimacy in relationships, shaping experiences of and expectations surrounding social support [27,28].

When forcibly displaced individuals and their families resettle to a permanent country of asylum, they disconnect almost overnight from the support they rely on from their family and community networks [29,30]. Research with Syrian families identified multi-faceted losses to social connections resulting from displacement. These losses and separation from friends and family members were interrelated with losses of identities and roles that were inherent to socio-centric notions of self and identity—e.g., as a caregiver to aging parents, provider, and protector for the family, neighbor, and engaged member of the community [26]. While advances in technology in the past decades facilitate retaining transnational connections, the help that people (women in particular) relied on to carry out daily activities and fulfill responsibilities constricts dramatically and suddenly as soon as they board the flight to the country of resettlement. This sudden loss of social support is part of a myriad of complex psychosocial processes at play among refugees in contexts of resettlement [26,31].

The need for various forms of social support is significant among refugee communities who are uprooted from home and exposed to trauma and chronic adversity. Findings from a scoping review of social support research with resettled refugees suggested that their needs for social support are diverse and nuanced [2]. Social support needs included commonly identified needs such as practical and emotional support as well as forms of support more directly tied to the challenges of re-establishing oneself in a new home space (e.g., companionship support, sense of belonging, and adjustment support) [2]. While some studies identified and measured the structural elements of social support (e.g., size, closeness, and frequency of contacts), others measured adequacy, satisfaction, and/or quantity (frequency) of received support, and still others focused on the perceived reliability of available support and the importance of source proximity [2]. Qualitative research has provided specific insights into women’s specific needs for social support in resettlement, including emotional support (confiding personal matters, providing comfort, and love), informational support (advice and information sharing), mentorship support (role modeling, guidance, coaching, and problem-solving), practical support (help with childcare, financial concerns, material needs, and transportation), relational support (feeling connected, a sense of belonging, and a sense of wholeness), and spiritual support, described as a sense of connection with God that offered meaning and comfort [2,32].

Social support is essential to well-being and integration in the resettlement context [29,30,33,34]. Resettled refugees contend with physical displacement from their countries and regions of origin, loss, and trauma associated with armed conflict and political instability, separation from family and community, complex asylum processes, and challenges of resettling in a new social environment [35,36]. In response, refugees seek to initiate familiar support systems as they negotiate the stressful life transitions during migration and early resettlement [33], yet this is challenging, particularly in resettlement contexts that foster individualism, self-sufficiency, and social isolation. Refugees originating from collectivist societies, especially, are known to depend on informal and implicit support systems while those from more individualistic Westernized societies depend more on formalized systems [29]. Informal sources include family (i.e., personal, surrogate, or transnational), churches or faith-based organizations, neighborhood resources, and/other immigrants; and formal sources of support may include professional service providers and social services, community centers, and ethnic organizations [2,30,32]. While informal and formal sources may lend themselves to providing some types of support over others, these distinctions can be fluid in resettlement as refugee clients lean on resettlement agencies for the support they would have normally sought from friends and family [33]. These shifts from relying on informal to formal sources of support are seen in other populations whose health statuses and/or life circumstances dramatically shift [37].

### 1.3. Best Practices in Scale Development

Best practices for developing and validating scales for health, social, and behavioral research articulated by Boateng and colleagues guided all aspects of this review [38]. These steps include domain identification (e.g., deductive or inductive or both), content validity judging for relevance and specificity, extraction of factors and test of dimensionality (e.g., exploratory factor analysis, confirmatory factor analysis), tests of reliability (e.g., internal consistency, test-retest reliability), and tests of validity (e.g., predictive, criterion, concurrent, convergent, discriminant, correlation, differentiation between known groups). These best practices are important to examine the processes and strategies that have been used in developing scales to determine whether existing social support scales are adequate and appropriate for use among refugees in resettlement. The steps identified as best practices also facilitate the assessment of the validity and reliability of the scales and determine whether adequate adaptation processes have been followed accordingly. More frequently, the adaptation processes used in most scales fall short of internal and external validity. Following best practices of scale development and validation, this paper will address any structural, reliability, and validity gaps in the literature.

### 1.4. Study Aims

A scoping review aims to broadly map an area of research to provide a comprehensive overview and analysis of the existing literature [39]. This analysis aims to examine the current application of existing social support scales in research with refugees in resettlement, assess their quality, and identify gaps in measurement to enhance research and practice.

## 2. Materials and Methods

The Arksey and O’Malley framework for conducting scoping reviews guided this review [39]. This framework distinguishes the purpose of scoping reviews as distinct from systematic and other types of comprehensive reviews that use systematic search and analytic strategies, although some methods overlap [39]. While there is no one way to conduct a scoping review, Arksey and O’Malley outline a clear approach to mapping areas of research in five steps: identifying the research question, identifying relevant studies, study selection, charting the data, and collating, summarizing, and reporting the results. Scoping reviews are effective in illuminating the scope, nature, and characteristics of studies, lending themselves to identifying gaps and summarizing and disseminating research findings in an accessible format; and in doing so, researchers, practitioners, and policy makers are positioned to make good use of the available evidence [39]. Following this multi-step framework, we formulated our research questions, identified relevant studies, selected studies for inclusion in the analysis, charted and analyzed the relevant data, and synthesized the results as illustrated in the steps described below.

### 2.1. Step 1: Formulating Research Questions

The following questions served as a rubric for this scoping review of instruments used in measuring social support among refugees in resettlement: (1) What standardized instruments have been used to measure social support among refugees in resettlement contexts? (2) To what degree were the instruments developed, adapted, and validated with refugees in resettlement? (3) To what extent do these scales have appropriate content and construct validity, generally and specifically, for resettled refugees? (4) What are the gaps in the measurement of social support among resettled refugees?

### 2.2. Step 2: Identifying Relevant Studies

A systematic approach was used to identify social support scales that have been used in research with refugees in resettlement. We developed a search strategy around two conceptual domains of social support and resettled refugees. Both controlled vocabulary terms and keywords were used to describe each of these domains. An initial search process using a detailed search string captured articles published up to June 2019 [(“Social support” OR “Informal support” OR “Emotional support” OR “Practical support” OR “Pragmatic support” OR “Informational support” OR “Instrumental support” OR “Appraisal support” OR “Social capital” OR “Social networks” OR “Psychosocial support” OR Isolation OR Belonging OR “Social connection” OR “Social engagement” OR “Social integration”) AND (Refugee*) AND (Resettle*) NOT qualitative] in eight databases: Web of Science, CINAHL, Sociological Abstracts, PubMed, Social Services Abstracts, Cochrane, PsycINFO, and Health and Psychosocial Instruments. To ensure that the search was comprehensive and up to date, we conducted a subsequent search using a refined search string (“social support” AND refugee AND resettle* NOT qualitative) for studies published between June 2019 and March 2023. All articles were exported and saved to Zotero. Searches were conducted by a team of students who were supervised by the second and third authors.

### 2.3. Step 3: Study Selection

Only peer-reviewed articles published in English were eligible for inclusion. Additionally, articles that met the following criteria were included: the study focused on refugees in resettlement; the study quantitatively measured perceived or received social support (e.g., as an independent, mediator, moderator, or dependent variable); and social support was measured using a multi-item instrument (minimum of two items). Articles were excluded if they did not meet any one of the inclusion criteria including if the study measured a construct adjacent to social support (e.g., social networks) and/or if the research team was unable to access or locate the full text. For this review, we used the United Nations definition that defines a refugee as a person who is unwilling or unable to return to his or her country of origin because of persecution or a well-founded fear of persecution based on race, religion, nationality, or membership in a particular social group or political opinions [40].

A total of 1185 titles of articles were screened for relevance in Zotero, and 423 were excluded for non-relevance. A total of 762 abstracts were then screened for eligibility, of which 661 were excluded for not including a quantitative measure of social support. A full-text review was conducted in 101 articles; of this number, 48 articles used instruments tangential to social support; 8 articles used individual indicators of social support; and 4 articles did not include refugees in resettlement contexts, resulting in the removal of 60 articles. We retained 41 articles in the final analysis, from which we identified a total of 17 distinct social support instruments used in research with resettled refugees, which form the basis for our review and analysis (Figure 1). Having identified the 17 social support instruments used in research with resettled refugees, we then searched for the full-length instruments and articles that described their original development and psychometric properties.

### 2.4. Step 4: Charting and Analyzing Data

We charted and analyzed data extracted from articles that described their development and original psychometric properties and their adaptation (as relevant) and psychometric properties in research with resettled refugee populations. To guide the charting and analysis process, we developed a standardized data extraction tool. The following information was extracted from the articles and charted in Excel: author, year of publication, country, study population and context, items and response options, domains/subscales, tests of validity, and tests of reliability. These data-points formed the basis of our analysis, informed by best practices for developing and validating scales [38].

## 3. Results

### 3.1. Social Support Instruments Identified in the Literature

Table 1 provides the complete list of social support instruments and the number of studies with resettled refugees that used those tools to measure social support. Of the seventeen total instruments, the three most common social support instruments used in studies with resettled refugees were the Multidimensional Scale of Perceived Social Support (MSPSS; *n* = 8) [41]; the ENRICHD Social Support Instrument (ESSI; *n* = 7) [42]; and the Social Provisions Scale (*n* = 4) [43]. The remaining fourteen instruments were each used in three or fewer studies identified in this search. Three social support instruments were developed for use with refugees or other immigrant or migrant populations. The Multisector Social Support Inventory (MSSI) was developed specifically for war-affected youth, and two recent instruments were developed specifically for resettled refugees: Building a New Life in Australia (BNLA) Social Support Scale [44] and the Refugee Social Support Inventory (RSSI) [45]. The remaining fourteen instruments were originally developed with or for university students, medical patients, and general community members.

### 3.2. Populations and Countries of Resettlement in Which Studies Were Conducted

The included studies were conducted with diverse samples of refugees from Afghanistan, Bhutan, Burma/Myanmar, Iraq, Nepal, Somalia, South Sudan, Vietnam, the former Soviet Union, and Syria, among other countries and regions of origin. The countries where the studies were conducted included Australia, Canada, Lebanon, Jordan, Norway, Sweden, Switzerland, and the U.S. The majority of studies were conducted in the U.S. (*n* = 25), followed by Norway (*n* = 4) and Canada (*n* = 3). All but three studies were conducted with adults (18 years or older).

### 3.3. Domains and Formats of the Instruments

Table 2 describes the key characteristics of the identified instruments. Most of the social support instruments used in research with resettled refugees consisted of two to six domains or subscales. Domains varied and were not consistent across scales. Emotional and practical social support subscales were the most frequently measured domains, followed by informational support as well as less common and more subtle forms of support that appeared to relate to building confidence and facilitating connection and belonging (e.g., affect, appraisal, belonging, social integration). Other domains appeared only once and appeared to capture forms of support that are derived from close relationships and that related to self-esteem and affirming one’s worth and value (e.g., attachment, reassurance of worth, reliable alliance, guidance, opportunity for nurturance, affection, positive interaction, support/cohesion, intimacy, affirmation, confidant support) as well as aid short/long term, duration, frequency of contact, availability, event-related support, adequacy, and satisfaction of social support. In three instances, subscales were formulated based on the source of support (e.g., the person providing the support), as is characteristic of the MSPSS, MSSI, and RSSI. The MSPSS grouped support items based on whether the support was coming from family, friends, or significant others; the MSSI grouped support based on whether it was coming from the nuclear family, extended family, peers, or adult mentors; and the RSSI grouped items based on whether support was coming from friends/relatives or concerned institutions. The number of items included in the measures ranged from five to twenty-five items. Eleven scales measured perceived social support, four measured received social support, and two measured both perceived and received social support (NSSQ and SSNI).

Most of the response scales for the instruments were formatted using a Likert scale, with only two using a binary response (“No or Yes”; “as much as I would like”; “much less than I would like”). Six scales used a five-point Likert scale; four of these ranged from one to five (e.g., “none of the time to all of the time”; no support to high support; never to most of the time) and two ranged from zero to four (e.g., “not at all to for a great deal”; never to almost always). Four scales adopted a four-point Likert scale, with three ranging from one to four (e.g., “strongly disagree to strongly agree” or “definitely false” to “definitely true”) and one ranging from zero to three (“wouldn’t know at all to would know very well”). Two adopted a seven-point Likert scale ranging from one to seven (e.g., “very strongly disagree to very strongly agree” or “almost never to almost always”) and one each of a three-point Likert scale (“not at all to a great deal”) and six-point Likert scale (“very satisfied to very dissatisfied”). In most cases, higher scores reflected greater social support, and lower scores reflected lower social support (perceived and/or received).

### 3.4. Psychometric Assessments of the Original Social Support Instruments

The validity of a scale is the extent to which an instrument measures the latent dimension or construct that was measured [54]. The reliability of a scale is the degree to which observed individual differences in scale items are indicative of true individual differences and the degree of consistency exhibited when a measurement is repeated under identical conditions [54]. Here, we describe the validity and reliability of the original social support instruments in terms of criterion validity (concurrent and predictive), convergent, discriminant, or divergent validity, comparisons between known groups, internal consistency, and test-retest reliability (Table 3), prior to any adaptation with and for resettled refugee populations.

Criterion validity is the degree to which there is a relationship between a given score of an instrument and performance with another measure of relevance [38,54]. There are two types of criterion validity: concurrent and concurrent predictive validity. Of the 17 scales used, only 5 reported assessing concurrent criterion validity, among which 2 used a latent construct that fits well with social support. Criterion predictive validity, the extent to which a measure predicts the answers to some other question or a result to which it ought to be related [38], was assessed in 15 out of the 17 scales. It was used as a predictor of depression anxiety, social integration, loneliness, marital adjustment, physical or emotional function, life stress, or sexual activity.

Convergent validity, the extent to which a construct measured in different ways yields similar results, was assessed in 10 out of the 17 scales. The social support instruments were measured against other related measures such as socially supportive behaviors, provision of social relations scale, self-esteem, social support, social network, social desirability, or other related indicators such as number of supportive persons, satisfaction with support, loneliness and family functioning, social contacts, and social activities.

Discriminant or divergent validity, the extent to which a measure is novel and not simply a reflection of some other construct [38,54], was assessed in half of the instruments against measures of depression, social network, social desirability, physical and mental health status, anxiety, self-help ideology, and reported satisfaction with the number of current friends. Differentiation and analyses by specific/known sub-groups examine whether the concept (e.g., social support) measured behaves as expected to specific sub-groupings based on theoretical and empirical knowledge and determine the distribution of the scale scores over the sub-groups [38,54]. This indicator of external validity was assessed in 11 out of the 17 scales. Gender or sex formed the basis of comparison. Additionally, age, minority status, ethnicity, marital status, socioeconomic status, and positive or negative self-statements were used. Generally, the instruments assessed at least three of the five markers of validity supporting psychometric validity.

The reliability of the instruments was assessed by tests for internal consistency using a Cronbach’s alpha coefficient (*n* = 17) and/or test-retest reliability using a Pearson correlations coefficient (*n* = 14). Best practice requires that scales have alpha coefficients greater than 0.70 as a minimum, with an optimal value ranging from 0.80 to 0.90 [38]. All the instruments except one had such optimal values. With regards to test-retest reliability, optimally the correlation coefficient should be greater than 0.70. Of the 14 scales that conducted the assessment, 9 had coefficients greater than 0.70, satisfying the rule of thumb.

### 3.5. Adaptation of Social Support Instruments in Studies with Resettled Refugees

Apart from translation, almost no substantial adaptations (e.g., changing wording of questions, adding or dropping items) to the instruments were reported in the forty-one studies where the scales were used among refugees in resettlement (Table 4). Twenty-six studies (63%) described written translations of social support instruments, and one reported oral translation. Only four (10%) reported any tests of validity, but twenty-four (59%) reported a test of reliability (internal consistency of scale items as per Cronbach’s alpha). By scale, five out of eight studies that used the MSPSS reported written translations and back-translations into Chin-Burmese, Arabic, and Nepali. One of these studies reported a careful process of reviewing the full survey, adapting or dropping specific questions, and working together to settle on oral translations of measures. Of the eight, one reported a test of validity and seven reported tests of reliability.

The ESSI was the second most used tool, used in six studies focused on Syrian refugees resettled in Sweden, Lebanon, and Norway. Only one study reported forward and back-translation, none reported psychometric evaluation or validation, and two reported tests of reliability. The Social Provisions Scale was used in four studies among refugees resettled in the U.S. and in Switzerland, of which only one reported translation and back-translation, none reported psychometric evaluation or validation, and three reported a test of internal consistency, which were above the minimum threshold of 0.70. The MOS SSS, MSSI, and Social Support Microsystems Scales were used in three studies each. The MOS SSS was used with Bhutanese refugees in the U.S. and Canada. In all three studies, forward and back-translation were conducted; however, no report was made of psychometric evaluation and validation, and only one study reported a measure of internal consistency.

The MSSI was used in studies with refugees from diverse countries of origin who resettled in the U.S. There was forward and backward translation reported in one study, and the assessment of reliability using Cronbach’s alpha in two of the three studies was above the recommended threshold of 0.70. The Social Support Microsystems Scale was used in three studies among refugees in the U.S. from Vietnam and the former Soviet Union. Of the three, two studies reported forward and back-translation, none reported validity tests, and two reported tests of reliability. Of the eleven remaining scales used in fourteen studies, forward and backward translation was reported in seven studies, psychometric evaluation and/or validation were reported in none, and a measure of internal consistency was assessed in six.

### 3.6. Assessments of Social Support Instruments Validated in Studies with Resettled Refugees

Our analysis identified four instruments as having been tested for validity to varying degrees with populations of resettled refugees. Here, we focused on assessing the psychometric evaluation process of the identified scales (Table 5).

Tonsing [55] evaluated the psychometric properties of the existing 12-item MSPSS [41] administered to Chin-Burmese (CB) refugees in the U.S. (also referred to as the CB-MSPSS). In this study, the MSPSS was translated and back-translated, an indication of content validity. With regards to construct validity, the authors started with an exploratory factor analysis, which supported the theoretical dimensions of the original MSPSS (friends, family, and significant other subscales), accounting for 82.15% of the total variance. This assessment was followed by a confirmatory factor analysis with the same sample of Chin-Burmese refugees, which showed moderate goodness of fit. The authors assessed for concurrent validity with a measure of psychological distress, which showed a significant negative intercorrelation. Tests of internal consistency of the three subscales were all high (0.94–0.96). Additional tests of predictive validity, convergent validity, discriminant validity, and comparison between known groups were not reported.

Gotvall et al. [68] evaluated the psychometrics of the existing seven-item unidimensional ESSI [42] with a sample of Syrian refugees who resettled in Sweden. Confirmatory factor analysis was used to test the scale’s dimensionality in this population. The use of a structural equation model (measurement model) indicated support for the scale’s unidimensionality. Measurement invariance was assessed to determine the equivalency of the scale by gender (male/female). Consistent with configural, metric, scalar, and strict invariance, the analysis produced a partial scalar invariance for the scale. The moderation and mediation models that included measures of torture exposure and post-traumatic stress disorder were indications of both predictive and discriminant validity. Tests of internal consistency produced an alpha of 0.91. Additional tests of comparison between known groups and concurrent validity were not reported.

Building a New Life in Australia (BNLA) Social Support Scale [44] was constructed using a pre-existing database from a cohort study with humanitarian migrants who resettled in Australia. Informed by the broader social support literature (not specific to resettled refugees), ten items were selected from a pre-existing dataset to create the scale in a secondary analysis. Exploratory factor analysis was used to assess construct validity, which produced two domains, with good to excellent internal consistency (0.86, 0.92). No other analyses to test the factor structure were reported. With regards to external validity, the analyses point to predictive validity, with psychological distress, and some form of discriminant validity, with gender and age groupings. Additional tests of content validity, concurrent validity, convergent validity, divergent validity, or measurement invariance were not reported.

The Refugee Social Support Inventory (RSSI) [45] was created as part of a study with Syrian refugee parents in Lebanon and Jordan. The RSSI is a 24-item measure used to assess the frequency of post-resettlement support received from families and institutions. The authors followed best practices by identifying items using both deductive (theoretical and empirical research) and inductive (observations and interviews) methods. Item identification was followed by an expert review of the items to determine the relevance and representation of items about refugee experiences in resettlement. It was not reported whether the population of interest was consulted in the expert review or whether cognitive interviews were conducted to complete the content validation process. Principal component analysis was conducted to establish construct validity, in which five domains were identified. Predictive validity with psychological distress was reported. A Cronbach’s alpha of 0.98 was reported for all items (not by domain). Additional tests of concurrent validity, convergent validity, discriminant/divergent validity, and comparisons between known groups were not reported.

## 4. Discussion

The purpose of this scoping review was to examine social support measurement in research with refugees in resettlement, assess the quality of existing instruments, and identify gaps to enhance research with this diverse population. Our final analysis included 41 studies conducted with refugees in diverse resettlement contexts that included a multi-item measure of social support. From these studies, we identified 17 standardized instruments of social support, 15 of which were not originally developed with refugees. Of the 15 not originally developed for refugees, only 2—the MSPSS and unidimensional ESSI (the most frequently used instruments)—were psychometrically evaluated to varying degrees with samples of resettled refugees, each in one study [55,68]. Our analysis also identified two recently published social support scales that specifically focused on the experiences of resettled refugees: the BNLA [44] and RSSI [45]. The RSSI was the only instrument out of the 17 identified in the current analysis that drew from formative work as part of the item identification process. Collectively, our analysis indicates the vast majority of research assessing social support among resettled refugees is conducted without tools created for or validated in the resettlement context, and many (41%) did not report internal consistency of social support scales used in research with resettled refugees.

Our analysis illuminates advancements in social support measurement among resettled refugees. Indications of progress include the efforts to psychometrically evaluate the MPSS and ESSI with resettled Chin-Burmese and Syrian refugees resettled in the U.S. and Sweden, respectively. It also includes the development of two scales specific to the context of resettlement, the BNLA with humanitarian migrants in Australia from diverse regions of origin and the RSSI with forcibly displaced Syrians living in Lebanon and Jordan. The CB-MSPSS, ESSI, BNLA, and RSSI show degrees of validity in the respective populations of resettled refugees in which they were psychometrically evaluated. The two scales that were not originally created for these populations (MSPSS and ESSI) are widely used and shown to be useful measures of social support writ large but are limited in their specificity and applicability to refugee populations in resettlement due to how and with what populations the items were originally developed. Although the MSPSS and ESSI seem to capture key facets of perceived social support in the general population that may be relevant to resettled refugees, they cannot be assessed as original social support scales for resettled refugees. Furthermore, items in the MSPSS and ESSI reflect perceptions of social support in very broad terms. In contrast, items from the BNLA and RSSI are specific to contexts of refugee resettlement. The BNLA, however, is limited by having been constructed based on the availability of variables in a pre-existing cohort study; in other words, the study was not designed to examine social support or create a measure of social support but did so after the fact. Furthermore, the BNLA is made up of items that measure level of knowledge in carrying out certain tasks in resettlement, raising questions about the degree to which it measures social support per se. The RSSI, in contrast, was intentionally created and validated to capture specific elements of social support relevant to refugees in resettlement; indeed, the process to develop the RSSI reflects best practices in developing original scales [38]. The extent to which the RSSI, developed in the context of Jordan and Lebanon, will perform in resettlement countries such as Australia, Canada, and the U.S. will need to be carefully evaluated moving forward.

While these studies indicate progress in social support measurement in populations of resettled refugees, our findings highlight the considerable gaps yet to be filled with regard to measurement development grounded in context-specific lived experiences and cross-cultural validation. Mainstream measures of social support that were originally developed with non-refugee populations in the Global North understandably fail to capture meanings, needs, and expectations of social support among people fleeing traumatic life experiences and resettling from diverse regions at a critical point in time when people separate from established networks and experience dramatic losses of social support. Notably absent from studies assessing the CB-MSPSS, ESSI, BNLA, and RSSI were tests of concurrent validity, convergent validity, and comparison between known groups and the use of cognitive interviews to enhance content validity. While nearly all of the studies relied on forward and backward translation methods as an assessment of content validity, many consider consensus and group-based methods of translation to be more robust and accurate [92,93]. Except for the RSSI, these scales in use do not readily lend themselves to nuanced and holistic measurements of a complex construct with meanings specific to refugee resettlement. None of the identified scales, moreover, incorporated a gendered perspective, which is critical in light of how cumulative losses of social support in forced migration disproportionately impact women whose household responsibilities depended heavily on support from family and community networks prior to resettling [29,33]. Nor did the studies incorporate a lifespan perspective, considering how priorities, needs, and expectations would be shaped by development and life stage. It is also important to consider whether the scales in use are set up to meaningfully measure changes in social support over time, which is an important consideration with regard to research, practice, and program evaluation [2]. Encouraging refugee-serving organizations to systematize assessments of social support in routine service delivery and programming, especially among women and other vulnerable client groups, would be an important step forward in addressing persistent gaps.

To address these gaps, we call for the prioritization of high-quality, context-specific, and culturally relevant measurements of social support in research with resettled refugees in recognition of how important social support is to almost every facet of life, especially health and well-being [94]. To conduct this, the field needs to psychometrically evaluate social support scales—ideally scales originally developed with the intended population—across populations of resettled refugees and resettlement contexts so they can be considered cross-culturally applicable. Finally, we need to invest time and resources into developing new social support instruments with resettled refugee populations, following best practices for scale development. This includes developing gender-specific scales as well as measures of social support for children, adolescents, and family systems [95]. We encourage researchers to carefully assess and select social support instruments based on contextual relevancy for resettled populations. Failing to do so limits research conclusions and does resettled groups a disservice in terms of being able to meaningfully inform the development of programs and practices that enhance social support post-resettlement [31].

### Limitations

The Arksey and O’Malley [39] five-stage framework guided the rigorous approach to our scoping review; however, this review is not without some limitations. Our identification of articles may have been limited in part because we only included articles published in English and those returned by our chosen search engines. Others may produce additional relevant articles, and articles may exist in other languages that are not captured in this review. Our analyses are based on what the articles reported, and we recognize that more may have been performed behind the scenes to enhance the validity and reliability of social support instruments. Finally, the 41 articles included in this review do not reflect the full breadth of the literature that measured social support among resettled refugees because we eliminated studies if they only used a one- or two-item measure of social support. Nonetheless, our systematic approach to this review provides a nuanced understanding of how social support is measured among refugees in resettlement.

## 5. Conclusions

While advances in social support measurement are emerging, this review highlighted the extent to which research with resettled refugees has overwhelmingly relied on social support scales that do not account for how war, culture, forced migration, and resettlement shape meanings of and needs for social support. This disconnect leads to missed opportunities in research and practice, so that social support gaps may be overlooked or misunderstood. The analysis highlights the need for rigorously developed social support scales that reflect the lived experiences of culturally diverse resettled refugees across the lifespan. The adaptation and psychometric evaluation of existing scales for new populations can be both an effective and efficient use of limited resources. Yet, the gold standard in measurement science is the development of original scales that account for lived experiences, cultural and linguistic nuances, and contextual factors. Measuring social support in this context requires meaningful engagement with those with relevant lived experiences and perspectives. While quantitative measurement of complex social constructs is fundamentally reductionist, we must strive for meaningful multidimensionality that lends itself to producing high-quality research on the one hand and informing practice on the other. To close this gap, our analysis highlights the need for rigorously developed social support scales that reflect the lived experiences, needs, and priorities of resettled refugees. Conducting such work requires a multi-stage, iterative process that begins with examining lived experiences and resulting social support needs from the perspective of individuals impacted by displacement and resettlement. Translating such insights into key areas, items, and domains for social support assessment requires mixed-method approaches and ongoing engagement with refugee communities to ensure that items are meaningful and understandable and reflect their experiences. Also, it calls for rigorous statistical modeling that explores the phenomenon as unidimensional, bidimensional, or multidimensional, followed by tests of both internal and external validity. Thus, a social support scale that captures the lived experiences, needs, and priorities of resettled refugees following our assessment is a significant and necessary next step, which will be useful for screening the needs of resettled refugees, evaluating existing support systems, informing policies on refugee integration, and assessing related social, economic, and health outcomes.

## Figures and Tables

**Figure 1 ijerph-21-00805-f001:**
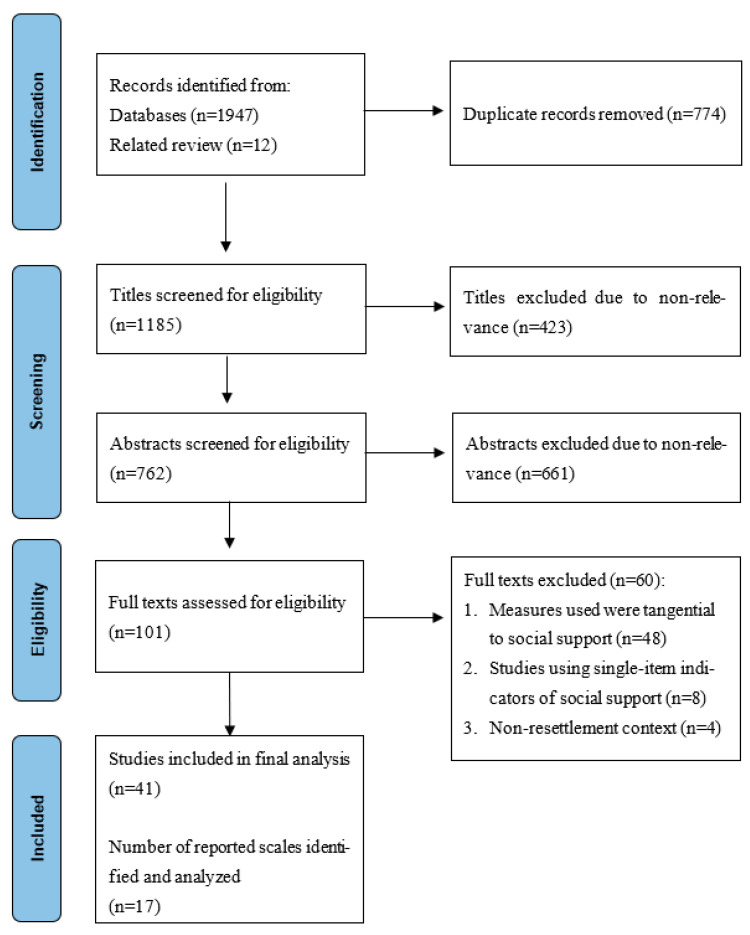
Flowchart of the search and selection process.

**Table 1 ijerph-21-00805-t001:** Social support instruments (*n* = 17) identified in studies with refugees in resettlement contexts (*n* = 41).

Instrument	# Studies
Multidimensional Scale of Perceived Social Support (MSPSS; [41])	8
ENRICHD Social Support Instrument (ESSI; [42])	7
Social Provisions Scale (SPS; [43])	4
Medical Outcomes Study Social Support Survey (MOS SSS; [8])	3
Multisector Social Support Inventory (MSSI; [46])	3
Social Support Microsystems Scales (SSMS; [4])	3
Interpersonal Support Evaluation Checklist (ISEL; [17])	2
Personal Resource Questionnaire ([47])	2
Norbeck Social Support Questionnaire (NSSQ; [48])	1
Duke–UNC Functional Social Support Questionnaire (FSSQ; [49])	1
Social Support Network Inventory ([50])	1
Refugee Social Support Inventory (RSSI; [45])	1
Building a New Life in Australia (BNLA) Social Support Scale (BNLA; [44])	1
Seeking social support subscale from the Ways of Coping Questionnaire ([51])	1
Social Support Questionnaire (SSQ6; [3])	1
Social Support Appraisal Scale ([52])	1
Perceived Social Support Scale—Friend Version (PSS-Fr; [7])	1

**Table 2 ijerph-21-00805-t002:** Characteristics of social support instruments used in studies with refugee populations in resettlement contexts (*n* = 17).

Instrument	Original # Items	Final # Items	Domains (# Subscales, Description)	Perceived/Received	Sources of Support
Multidimensional Scale of Perceived Social Support (MSPSS; [41])	24	12	3, by source	Perceived	Family, friends, and significant other
ENRICHD Social Support Instrument (ESSI; [42])	7	7	None specified	Perceived	1 item asks if currently married or living with a partner/spouse
Social Provisions Scale [43]	12	24	6, attachment, social integration, reassurance of worth, reliable alliance, guidance, opportunity for nurturance	Perceived	None specified
Medical Outcomes Study Social Support Survey (MOS SSS; [8])	50	19	5, emotional, informational, affection, tangible, positive interaction	Perceived	None specified
Multisector Social Support Inventory (MSSI; [46])	15	11	4, by source	Perceived	Nuclear family, extended family, peer support, adult mentor support
Social Support Microsystems Scales [4]	28	21	3, support/cohesion, daily hassles, involvement	Perceived	Family, peer, school, neighborhood context
Interpersonal Support Evaluation Checklist (ISEL-12 items; [53])	20	12	3, appraisal, belonging, tangible	Perceived	Close confidant
Personal Resource Questionnaire [47]	25	25	5, intimacy, social integration, nurturance, worth, assistance	Perceived	None specified (specified in part 2)
Norbeck Social Support Questionnaire (NSSQ; [48])	9	9	6, affect, affirmation, aid short term/long term, duration, frequency of contact, recent loss	Perceived and Received	Network members
Duke-UNC Functional Social Support Questionnaire (FSSQ; [49])	14	8	2, affective and confidant support	Perceived	Confidant
Social Support Network Inventory [50]	20	11	5, availability, practical help, reciprocity, emotional, event-related	Perceived and Received	None specified
Refugee Social Support Inventory (RSSI; [45])	24	24	5, informational and instrumental support from institutions and instrumental, informational, and emotional support from family	Received	Family/relatives, institutions
Building a New Life in Australia (BNLA) Social Support Scale [44]	10	10	2, emotional/instrumental, informational	Received	National or ethnic community, religious community, community groups
Seeking social support subscale from the Ways of Coping Questionnaire [51]	6	6	None specified	Received (sought)	Relative/friend, professional help
Social Support Questionnaire (SSQ6; [3])	27	27, 6	3, appraisal, belonging, tangible	Perceived (satisfaction)	None specified
Social Support Appraisal Scale [52]	23	23	2, adequacy, satisfaction	Perceived	Friends and family relations
Perceived Social Support Scale—Friend Version (PSS-Fr; [7])	20	20	2, perceived social support (from friends)	Perceived	Friends

**Table 3 ijerph-21-00805-t003:** Tests of validity and reliability of original social support instruments (not developed specifically for refugees in resettlement).

		Reported Tests of Validity					Reported Tests of Reliability	
Instrument	Sample	Criterion Concurrent	Criterion Predictive	Convergent	Discriminant or Divergent	Comparison between Known Groups	Internal Consistency	Test-Retest Reliability
Multidimensional Scale of Perceived Social Support (MSPSS; [41])	275 undergraduate students (136 female, 139 male; 17–22 years old)	Not reported	Depression, anxiety	Not reported	Depression and anxiety	Gender differences	α = 0.91, 0.87, 0.85 (SO, family, friends); composite: 0.88	*r* = 0.72, r = 0.85, 0.75; composite: 0.85
ENRICHD Social Support Instrument (ESSI; [42])	196 patients after myocardial infarction (121 male, 74 female, 67% White)	Perceived social support	Mortality in cardiovascular patients	Inventory of socially supportive behaviors	Social network questionnaire	Gender, minority status	α = 0.86	Not reported
Social Provisions Scale [43]	1792 (1183 university students, 303 teachers, 306 nurses)	Not reported	Loneliness, social integration, satisfaction with friends, depression, physical health	Satisfaction with support, attitude towards support, # of supportive persons, # of helping behaviors	Social desirability, depression, extraversion, neuroticism	Sex, age	α = 0.65–0.76; composite: 0.915	Multiple studies reported reliability analyses
Medical Outcomes Study Social Support Survey (MOS SSS; [8])	2987 patients (39% male, 80% White, aged 18–98)	Not reported	Loneliness, emotional ties, physical functioning and pain intensity	Loneliness and family functioning	Physical and mental health status	Not reported	Emotional: α = 0.96; tangible α = 0.92; positive α = 0.94; affection α = 0.91; composite: 0.97	r = 0.72; tangible r = 0.74; positive r = 0.72; affection r = 0.76; emotional composite: 0.78
Multisector Social Support Inventory (MSSI; [46])	618 Sarajevo secondary school students (30% male, 63% female)	Perceived social support	Depression	Provision of social relations scale	Anxiety, somatic distress, grief, post-traumatic stress	Not reported	α = 0.80–0.93	r = 0.59–0.81
Social Support Microsystems Scales [4]	988 youth (59% female, 26% Black, 26% White, 37% Latinos; low SES)	Not reported	Sexual activity	Not reported	Not reported	Gender, ethnicity	α = 0.74–0.81	r = 0.48, r = 0.40, and r = 0.35 over 10 months
Interpersonal Support Evaluation Checklist (ISEL-12 items; [53])	Multiple samples (mostly college students)	Not reported	Psychological symptomatology—depression scale; physical symptomatology	Inventory of socially supportive behaviors; emotional support subscales; Rosenberg self-esteem scale	Social desirability; social anxiety	Not reported	α = 0.77–0.86; composite: 0.88–0.90	r = 0.82–0.87 over 6 weeks to six months; 6 months—0.45–0.72
Personal Resource Questionnaire [47]	149 white middle-class spouses of individuals with multiple sclerosis	Not reported	Marital adjustment and family functioning	Not reported	Self-help ideology	Not reported	α = 0.89	r = 0.61–0.77
Norbeck Social Support Questionnaire (NSSQ; [48])	135 graduate and undergraduate nursing students (128 female, 7 male)	Not reported	Psychiatric symptomatology, life stress	Social support questionnaire (Cohen/Lazarus)	Not reported	Age, educational level, ethnic background	α = 0.95–0.98	r = 0.85–0.92; r = 0.88–0.96
Duke-UNC Functional Social Support Questionnaire (FSSQ; [49])	401 patients (28% Black, 72% White, 78% female, 22% male, 18+ years old)	Not reported	Physical function, emotional function, and symptom status	Social contacts and social activities questionnaire	Not reported	Sex, marital status, age, race, employment status, education, SES, and living situation	α = 0.52–0.73	r = 0.66 over 13.1 days, range of 6 to 30 days
Social Support Network Inventory [50]	207 nonpatients (100 students, 74 from an urban community, 32 members of a religious commune)	Not reported	Unipolar depression, social adjustment scale, Hamilton depression rating scale	Unipolar patient–clinician ratings	Not reported	Network of support, age, sex, and SES	Kuder–Richardson-20: α = 0.821, 0.76–0.91	r = 0.87, over two weeks
Seeking social support subscale from the Ways of Coping Questionnaire [51]	83 psychiatric outpatients, 62 spouses of patients with Alzheimer’s disease, 425 medical students	Not reported	Appraisal, anxiety, depression, and distress	Not reported	Not reported	Gender	0.75—medical students, 0.79—spouse of patient, 0.81—psychiatric outpatients	Not reported
Social Support Questionnaire (SSQ6; [3])	182 undergraduate students (108 female, 74 male)	Not reported	Anxiety, depression, hostility, social competence, loneliness	Inventory of socially supportive behaviors, social network list, family environment	Not reported	Not reported	α = 0.75, α = 0.79	r = 0.44, 0.39
Social Support Appraisal Scale [52]	5 university student samples (N = 517); 5 community samples (N = 462)	Not reported	Psychological distress, CES-depression scale, loneliness, negative affect	Social support questionnaire; provision of social relations scale; revised Kaplan scale; network orientation scale	Reported satisfaction with number of current friends; satisfaction with the quality of current friends; feeling different from friends	Not reported	α = 0.90, α = 80, α = 0.84	Multiple studies reported reliability analyses
Perceived Social Support Scale—Friend Version (PSS-Fr; [7])	Study 1: 222 undergraduate students; Study 2: 105 undergraduate students	Not reported	Life stress, symptomatology, depression, psychasthenia, schizophrenia	Social desirability; social network	Not reported	Positive and negative self-statement groups	α = 0.88, α = 0.90	r = 0.83 over 1 month

**Table 4 ijerph-21-00805-t004:** Tests of validity and reliability of social support instruments in studies conducted with refugee populations in resettlement contexts (*n* = 41).

Instrument	Study Citation	Population and Resettlement Country	Sample	Translation	Tests of Validity	Tests of Reliability
Multidimensional Scale of Perceived Social Support (MSPSS; [41])	[55]	Burmese refugees in the U.S.	242 adults (18+); 53.3% male	English into Chin-Burmese, two bilingual translators, translated, back-translated	Construct validity, concurrent/predictive validity	Cronbach’s alpha = 0.96 friends, 0.94 family, 0.96 significant other
	[56]	Syrian refugees in Canada	1924 adults (18+); 48.8% males; 51.2% females	Arabic translation and back-translation by two bilingual Syrian Canadians	Not reported	Cronbach’s alpha = 0.87
	[57]	Bhutanese refugees in the U.S.	44: adults (18+); 68.1% female	Back-translation by one bilingual Nepali and two bilingual Bhutanese	Not reported	Cronbach’s alpha = 0.92
	[58]	Chin-Burmese refugees in the U.S.	242: adults (18+) 53% male, 79% female	Back-translation by bilingual Chin-Burmese translators	Not reported	Cronbach’s alpha = 0.94
	[59]	Syrian refugees and asylum seekers in Switzerland	15: families (8+)	Not reported	Not reported	Not reported
	[60]	Bhutanese refugees in the U.S.	225 adults (18+), 113 male, 112 female	Not reported	Not reported	Cronbach’s alpha = 0.92
	[61]	Bhutanese refugees in Australia	148 adults (18+), 51.4% male, 48.6 female	Nepali nationally accredited translator and bilingual translators	Not reported	Cronbach’s alpha = 0.80–0.93
	[62]	Refugees in the U.S. from the Great Lakes region of Africa	36 adults (18+), 19 female, 17 male	Oral translations	Not reported	Cronbach’s alpha = 0.82–0.94
ENRICHD Social Support Instrument (ESSI; [42])	[63]	Syrian refugees in Sweden	1215 adults (18+); 475 male, 250 female	Not reported	Not reported	Not reported
	[64]	Syrian refugees in Lebanon and Norway	353 adults (18+); 181 female, 171 male	Not reported	Not reported	Not reported
	[65]	Syrian refugees in Lebanon and Norway	353 adults (18+); 181 female, 171 male	2 independent forward translations, reconciliation of forward translations, back-translation, harmonization	Not reported	Cronbach’s alpha = 0.85
	[66]	Syrian refugees in Norway	902 adults (18+); 35.5% female, 64.5% male	Not reported	Not reported	Not reported
	[67]	Syrian refugees in Sweden	1215 adults (18+)	Not reported	Not reported	Not reported
	[68]	Syrian refugees in Norway	1215 adults (18+); 475 male, 250 female	Not reported	Construct validity, comparison between known groups	Cronbach’s alpha = 0.906
Social Provisions Scale [43]	[69]	Refugees in the U.S. from east and central Africa	97 adults (18+)	Not reported	Not reported	Cronbach’s alpha = 0.915
	[70]	Bhutanese refugees in the U.S.	200 adults (18+)	Not reported	Not reported	Not reported
	[71]	Afghan refugees in the U.S.	49 adults (18+), 41% male, 59% female	Not reported	Not reported	Cronbach’s alpha = 0.96
	[72]	Refugees from multiple countries of origin in Switzerland	94 adults (18+), 85.1% male	Translated by native speakers or trained translators and back-translated in blind written form	Not reported	Cronbach’s alpha = 0.77
Medical Outcomes Study Social Support Survey (MOS SSS; [8])	[73]	Bhutanese refugees in the U.S. and Canada	190 older adults (50+)	Forward- and back-translated from English to Nepali by Bhutanese team members	Not reported	Cronbach’s alpha = 0.89
	[74]	Bhutanese in the U.S.	50 adults (18+), 62% female	Forward and back-translation into Nepali	Not reported	Not reported
	[75]	Bhutanese refugees in the U.S.	65 adults (18+), all female	Translated to Nepali by a professional translation service	Not reported	Not reported
Multisector Social Support Inventory (MSSI; [46])	[76]	Refugees in the U.S. from Afghanistan, Great Lakes region in Africa, Iraq, Syria	290 adults (18+); 52% female	Translated and back-translated from English into Arabic, Dari, French, Kiswahili, and Pashto	Not reported	Cronbach’s alpha = 0.85–0.89 family, 0.91–0.93 ethnic community, 0.88–0.90 non-ethnic community
	[77]	Refugees in the U.S. from Afghanistan, Great Lakes region in Africa, Iraq, and Syria	178 adults (18+), 46.6% male	Not reported	Not reported	Cronbach’s alpha = 0.905
	[78]	Refugees in the U.S. from Burma, Cameroon, Liberia, Uganda	84 HIV-positive adults (18+), 26 refugees	Not reported	Not reported	Not reported
Social Support Microsystems Scales [4]	[79]	Vietnamese refugees in the U.S.	203 adults (18+)	Translated into Vietnamese and back-translated into English	Not reported	Not reported
	[80]	Vietnamese refugees in the U.S.	212 adults (18+), 103 females and 108 males	Translated into Vietnamese and back-translated by a different set of translators	Not reported	Cronbach’s alpha = 0.86 spouse, 0.83 other family, 0.75 Vietnamese friends, 0.79 American friends
	[81]	Refugees from the former Soviet Union in the U.S.	110 adolescents	Not reported	Not reported	Cronbach’s alpha = 0.76
Interpersonal Support Evaluation Checklist (ISEL-12 items; [53])	[82]	Iraqi migrants and refugees in the U.S.	298 newly arrived, average age: 33.4 years	Translated by an Arabic bilingual psychiatrist and back-translated to English by a separate language expert	Not reported	Cronbach’s alpha = 0.89
	[83]	Kosovar refugees in the U.S.	61 adults (18+), 38 male, 48 female	Not reported	Not reported	Cronbach’s alpha = 0.67
Personal Resource Questionnaire [47]	[84]	Pregnant asylum seekers, refugees, and non-refugee immigrants in Canada	774 adults (18+), all female	Translated and back-translated into 13 languages and pretested with monolingual individuals to assess the clarity of the questions in each language	Not reported	Not reported
	[85]	Somali and South Sudanese refugees in Canada	58 adults (18+), 31 male, 27 female	Translated into Arabic, Sudanese, and Somali	Not reported	Not reported
Norbeck Social Support Questionnaire (NSSQ; [48])	[86]	Bhutanese refugee in the U.S.	45 adults (18+), all female	Written at a 5.5-grade English reading level then translated to Nepali; back-translated by Nepali-speaking staff	Not reported	Not reported
Duke-UNC Functional Social Support Questionnaire (FSSQ; [49])	[87]	Unaccompanied refugee minors in Norway	95 adolescents (<16), 76 male, 19 female	Not reported	Not reported	Cronbach’s alpha = 0.86
Social Support Network Inventory [50]	[86]	Bhutanese refugees in the U.S.	45 adults (18+), all women	Translated and back-translated	Not reported	Not reported
	[88]	Bhutanese in the U.S.	112 adults	Translated into Nepali by a Bhutanese professional interpreter and back-translated by another community leader	Not reported	Cronbach’s alpha = 0.77
Refugee Social Support Inventory (RSSI; [45])	[45]	Syrian refugees in Lebanon and Jordan	1000 parents (456 fathers, 544 mothers, 25–67 years old)	Original scale created for refugees presumably in Arabic, no report on translation processes	Content, construct, and predictive validity	Cronbach’s alpha = 0.98
Building a New Life in Australia (BNLA) Social Support Scale [44]	[44]	Humanitarian migrants in Australia from Africa, Middle East, southeast, central, and southern Asia	2264 adults (18+); 55.2% men, 44.8% women	Items were translated into multiple languages (e.g., Arabic, Persian, Dari) in the original surveys	Construct validity	Cronbach’s alpha = 0.80
Seeking social support in Ways of Coping [51]	[89]	Somali refugees in the U.S.	65 women	Translated and back-translated by skilled Somali study staff and external readers.	Not reported	Not reported
Social Support Questionnaire (SSQ6; [3])	[90]	Refugees/asylum seekers in the U.S. from Middle East, Africa, Europe, Asia, and South America	48 adults (18+), 27 women, 20 men	No written translations	Not reported	Not reported
Social Support Appraisal Scale [52]	[91]	South Sudanese refugees in the U.S.	172 adults (18+), all men	Translated into Arabic by a Sudanese professional and reviewed for accuracy	Not reported	Not reported
Perceived Social Support Scale—Friend Version (PSS-Fr; [7])	[7]	Refugees from the former Soviet Union in the U.S.	226 adolescents	Not reported	Not reported	Cronbach’s alpha = 0.67

**Table 5 ijerph-21-00805-t005:** Overview of four social support scales validated to varying degrees with refugee populations in the context of resettlement.

	CB-MSPSS [55]	ESSI [42]	BNLA Social Support Scale [44]	RSSI [45]
Dimensionality	Multidimensional Perceived availability of support from friends (four items) Perceived availability of support from family (four items) Perceived availability of support from significant other (four items)	Unidimensional scale of perceived support (seven items)	Multidimensional Emotional/instrumental support received from national/ethnic community, religious community, other community groups (three items) Informational support received (seven items) (measured by level of knowledge of how to complete key tasks)	Multidimensional Instrumental support received from family (six items) Informational support received from family (four items) Emotional support received from family (three items) Instrumental support received from institutions (six items) Informational support received from institutions (five items)
Original sample	College students [41]	Patients after myocardial infarction [42]	Humanitarian migrants in Australia from Africa, Middle East, southeast/central/south Asia (items identified deductively in a secondary analysis)	Syrian refugees in Lebanon and Jordan (deductive and inductive approaches to item identification)
Validation sample	Chin-Burmese refugees in U.S.	Syrian refuges in Sweden		
Assessments of validity	Indications of face validity, construct validity, concurrent validity	Indications of predictive validity, discriminant validity	Indications of construct validity, predictive validity, discriminant validity	Indications of face validity, content validity, construct validity, predictive validity
Assessments of reliability	Excellent internal consistency	Excellent internal consistency	Good to excellent internal consistency by subscale	Excellent internal consistency (reported on all items)
Sample items	My friends really try to help me; I obtain the emotional help and support I need from my family; I have a special person who is a real source of comfort to me.	Is there someone available to whom you can count on to listen to you when you need to talk? Is there someone available to you to give you good advice about a problem? Is there someone available to help with daily chores?	Given support/comfort from national or ethnic community; know how to look for job; know how to use public transport	Did anyone from your family or relatives give you a place to stay? Did anyone from your family or your relatives express interest and concern in your well-being or psychological condition? Did anyone from the concerned institutions help you to secure clothing?
Response options	Very strongly disagree; strongly disagree; mildly disagree; neutral; mildly agree; strongly agree; very strongly agree	None of the time; a little of the time; some of the time; most of the time; all of the time	Emotional/instrumental support: no, sometimes, yes; informational support: would not know at all, would know a little, would know fairly well, would know very well	Never; rarely; sometimes; most of the time

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
