# Peer review of "A Scoping Review of Instruments Used in Measuring Social Support among Refugees in Resettlement"

_ijerph, 2024, doi:10.3390/ijerph21060805_

Round 1

Reviewer 1 Report

Comments and Suggestions for Authors

Author Response

Reviewer 1 Feedback & Author Responses

  1. Overall the manuscript is very well written and addresses a need in the published literature.

Response 1: Many thanks for your positive feedback.

  1. Literature on Social Support Measurement and Measuring Social Support among Refugees During Resettlement are nicely summarized.

Response 2: Thank you.

  1. Misplaced periods on line 173 and 340.

Response 3: Thank you for pointing this out – we corrected it.

  1. Line 308 and table 3 column: Is there another way to state or describe “comparisons between known groups”? Does this mean the specified groups in the column were analyzed? The meaning and relevance is unclear in this section dealing with psychometric evaluation. Perhaps “Compares and Analyzes Specific Subgroups” would be clearer.

Response 4: Thank you. This has been addressed

  1. Section 3.5: I suggest breaking this into more than one paragraph.

Response 5: Thank you for your suggestion. We broke this section into 3 paragraphs.

  1. The discussion section is focused on research implications. Gaps (see line 503) could include social support assessment as a policy requirement for resettlement agencies.

Response 6: Thank you for the suggestion. We added this to the discussion, where and as suggested.

Reviewer 2 Report

Comments and Suggestions for Authors

Dear authors,

I appreciate that I got a chance to review your interesting work. The review is easy to read and transparent. I only have some minor points for you to consider.

First, I would have liked to see a discussion somewhere on the specific aspect of the applicability of the scales in western/non-western contexts. For example, Gottlieb and Bergen (2010) claim that ESSI is a good instrument for measuring social support in non-western communities. It would have been helpful for me to understand that assertion in relation to: 1) the RSSI and how it was constructed, and 2) the lack of adaptations that you write about in 3.5.

I also think that you could hone the conclusions a bit, especially the last sentence. Whereas I agree with you and the sentence is well supported by your work, it is a statement that could be made on page 1 as well. Your review shows you have more to say. 

Finally, I have a small comment on the format of numbers used, which is sometimes a little confusing. Since references are written as a number in parentheses in this journal I think it would be great to be consistent. On lines 68 and 93 you write the year in parentheses. I guess that is used to point to the fact that there is a ten year gap between the different Gottlieb-papers. I think it is better to write it without parentheses or take it out. Further, in 2. and 2.2. you write numbers that are not references in parentheses. In 2.1., on the other hand you write them with only a single parenthesis, like this 1). That is a clearer way of writing it, at least to me, and avoids confusion with the references.

I wish you all the best and look forward to reading your future work as well.

Author Response

  1. I appreciate that I got a chance to review your interesting work. The review is easy to read and transparent. I only have some minor points for you to consider.

Response 1: Many thanks for your positive feedback.

  1. First, I would have liked to see a discussion somewhere on the specific aspect of the applicability of the scales in western/non-western contexts. For example, Gottlieb and Bergen (2010) claim that ESSI is a good instrument for measuring social support in non-western communities. It would have been helpful for me to understand that assertion in relation to: 1) the RSSI and how it was constructed, and 2) the lack of adaptations that you write about in 3.5.

Response 2: We appreciate the feedback. The issues of whether the scales, including the ESSI, are applicable in western/non-western contexts may not be warranted in this analysis because our population of concern is refugees in contexts of resettlement. The ESSI may indeed effectively measure aspects of social support that are to some degree universal across cultures and contexts. However, these broad (often vague) notions of social support fail to capture the true picture and many of the salient needs for social support that are specific to the lived experiences and outcomes of refugees following war, forced displacement, and resettlement. In contrast to the ESSI, the RSSI was originally and specifically developed to measure social support among refugees in a resettlement setting. We address some of these issues in the discussion.

  1. I also think that you could hone the conclusions a bit, especially the last sentence. Whereas I agree with you and the sentence is well supported by your work, it is a statement that could be made on page 1 as well. Your review shows you have more to say. 

Response 3:  Thank you for this suggestion. We have elaborated the discussion, highlighting some of the different approaches and methods required to develop social support scales.

  1. Finally, I have a small comment on the format of numbers used, which is sometimes a little confusing. Since references are written as a number in parentheses in this journal I think it would be great to be consistent. On lines 68 and 93 you write the year in parentheses. I guess that is used to point to the fact that there is a ten year gap between the different Gottlieb-papers. I think it is better to write it without parentheses or take it out. Further, in 2. and 2.2. you write numbers that are not references in parentheses. In 2.1., on the other hand you write them with only a single parenthesis, like this 1). That is a clearer way of writing it, at least to me, and avoids confusion with the references.

Response 4: Thank you for pointing this out. We included the years without parentheses. We removed the numbers in 2, and we checked the document for consistency with how any numbers (and letters) appear.

  1. I wish you all the best and look forward to reading your future work as well.

Response 5: Many thanks for your feedback and helpful suggestions.

Reviewer 3 Report

Comments and Suggestions for Authors

Dear authors, I appreciated the opportunity to review this thoughtful work. Your approach is comprehensive and I believe this offers a valuable contribution to the literature. There are just a few potential revisions I would like to offer for your consideration.

1) You describe how measuring social support using existing instruments that have not been validated in refugee populations pose a challenge for truly understanding social support needs. Could length of time since resettlement, length of time as an asylum seeker, and other factors play a role in the suitability of existing instruments? Perhaps include this information in your review of existing instruments.

2) Related to this, more information about the unique social support needs of refugees would enhance the introduction.

3) More information about the Arksey and O'Malley framework would be helpful to better understand your approach. 

Thank you again for your work!

Author Response

I appreciated the opportunity to review this thoughtful work. Your approach is comprehensive and I believe this offers a valuable contribution to the literature. There are just a few potential revisions I would like to offer for your consideration.

Response: Many thanks for your feedback and review.

1) You describe how measuring social support using existing instruments that have not been validated in refugee populations pose a challenge for truly understanding social support needs. Could length of time since resettlement, length of time as an asylum seeker, and other factors play a role in the suitability of existing instruments? Perhaps include this information in your review of existing instruments.

Response 1: We agree with the reviewer that time is a very important factor in considering people’s social support following resettlement. This speaks to the need for longitudinal research that measures social support over time, and for a social support measure that is sensitive to the specific needs immediately post-resettlement and that ideally can be used over time, as well. We’re unable to include this information in our review of the existing instruments because it wasn’t a consideration in the articles that we reviewed (in terms of scale validation).

2) Related to this, more information about the unique social support needs of refugees would enhance the introduction.

Response 2: Thanks for this suggestion. We have added to this section, summarizing findings from relevant literature that demonstrates the diverse and varied needs of refugees including those of women and other vulnerable refugee communities.

3) More information about the Arksey and O'Malley framework would be helpful to better understand your approach. 

Response 3: Thank you for this suggestion. We added some information on the framework in the first paragraph of the materials and methods section.  

Thank you again for your work!

Response: We thank you for your review and encouragement.